# Interpretable multi-timescale models for predicting fMRI responses to continuous natural speech

**Shailee Jain**
Department of Computer Science
The University of Texas at Austin
Austin, TX 78712
shailee@cs.utexas.edu

**Vy A. Vo**
Brain-Inspired Computing Lab
Intel Labs
Hillsboro, OR 97124
vy.vo@intel.com

**Shivangi Mahto**
Department of Computer Science
The University of Texas at Austin
Austin, TX 78712
shivangi@utexas.edu

**Amanda LeBel** *
Department of Neuroscience
The University of Texas at Austin
Austin, TX 78712
amanda.lebel@berkeley.edu

**Javier S. Turek**
Brain-Inspired Computing Lab
Intel Labs
Hillsboro, OR 97124
javier.turek@intel.com

**Alexander G. Huth**
Computer Science & Neuroscience
The University of Texas at Austin
Austin, TX 78712
huth@cs.utexas.edu

## Abstract

Natural language contains information at multiple timescales. To understand how the human brain represents this information, one approach is to build encoding models that predict fMRI responses to natural language using representations extracted from neural network language models (LMs). However, these LM-derived representations do not explicitly separate information at different timescales, making it difficult to interpret the encoding models. In this work we construct interpretable multi-timescale representations by forcing individual units in an LSTM LM to integrate information over specific temporal scales. This allows us to explicitly and directly map the timescale of information encoded by each individual fMRI voxel. Further, the standard fMRI encoding procedure does not account for varying temporal properties in the encoding features. We modify the procedure so that it can capture both short- and long-timescale information. This approach outperforms other encoding models, particularly for voxels that represent long-timescale information. It also provides a finer-grained map of timescale information in the human language pathway. This serves as a framework for future work investigating temporal hierarchies across artificial and biological language systems.

## 1   Introduction

Natural language contains information at multiple timescales, ranging from phonemes to narratives [1]. The human brain processes language using a hierarchy of representations at different timescales [2, 3]. Early stages represent acoustic and word information at the sub-second scales, while at later stages information is combined over many seconds to derive meaning [3, 4]. These representations have

been mapped using fMRI responses to language that is scrambled at paragraph, sentence, and word scales [2, 3]. Short timescale areas respond to all these stimuli, while long timescale areas respond weakly to language scrambled at the word or sentence scale. This method has been effective at revealing where these representations are in the brain, but is unable to reveal how these representations are computed or what information is contained in each representation beyond its timescale.

One approach for studying how the brain computes language representations is encoding models [5–7]. These models predict fMRI responses to language using features extracted from the language stimuli. One method for extracting highly effective features is to use pre-trained language models (LMs) [8, 9]. This method can also be used to investigate representational timescales by manipulating the number of words passed to LM, or the *context length* [8, 9]. Separate encoding models are built using features extracted with different context lengths, and then the timescale of each brain area is determined by comparing prediction performance of these models [8]. Yet the results of this method can be difficult to interpret and susceptible to confounds. Further, rather than exploiting different timescale representations within the LM, this method simply ablates the input, potentially forcing the LM to produce off-manifold behavior.

An alternative to manipulating context length is to construct LMs where different timescales are explicitly represented by different units. One promising approach is offered by Tallec et al. [10], who showed that the forget gate bias in a long short-term memory (LSTM) unit determines its timescale. Positive bias implies slow 'forgetting' and a long timescale, while negative bias implies a short timescale. Building on this theoretical work, we construct an explicitly multi-timescale LSTM LM where the forget gate biases are fixed, giving each unit a defined timescale [11]. The distribution of timescales is selected to match known statistical properties of language [1]. Since each unit in the multi-timescale LM has an explicit timescale, we can use the encoding model weights on each unit to directly estimate the timescale of a voxel without manipulating the input context length. This allows us to create a finer-grained, more interpretable map of timescale representations during natural language comprehension across human cortex.

We also introduce a new method for creating encoding model features from LSTM representations that vary at different timescales. Any fMRI encoding model must transform the time domain of the representations to the time domain of the slow blood-oxygen-level dependent (BOLD) signal. This is straightforward for representations with uniform temporal properties [12]. However, the representation of a short-timescale unit will vary rapidly over time, while a long-timescale unit varies slowly over time. To accurately transform these representation to the time domain of the BOLD signal, we introduce an interpolation method that helps maintain the temporal properties of the LSTM representations. Taken together, these methodological contributions enable the multi-timescale LSTM to assign voxel timescales in a more accurate and interpretable way, paving the way for further investigation into how timescale information is extracted and represented in natural language.

## 2 Language encoding models for fMRI

### 2.1 Natural speech fMRI experiment

To build encoding models for language, we used data from an fMRI experiment comprising 6 human subjects (3 female) listening to spoken narrative stories from *The Moth Radio Hour* (an English language podcast) [13]. These rich, complex stimuli are highly representative of language that humans encounter on a daily basis. Understanding each 10-15 minute story requires subjects to integrate information across thousands of words. The 27 stories (26 train, 1 test) consisted of ∼57,900 total words (∼5,200 unique words). Each story was transcribed and the transcript was aligned to the audio [14] to find the exact time each word was spoken. All subjects were healthy with normal hearing, and gave written informed consent. The experimental procedure was approved by the local Institutional Review Board. Whole-brain MRI data was collected every 2 seconds. MRI acquistion details can be found in Supplementary Section 3.

### 2.2 Voxel-wise encoding models

The encoding model framework is shown in Figure 1. Encoding models learn to approximate the function $f(S) \to R$ that maps the stimulus $S$ to the elicited brain response $R$. Here, $S$ is a sequence of word tokens $w_1, w_2, \ldots, w_{n_w}$, and $R$ is the BOLD timeseries. $f(S)$ is often assumed

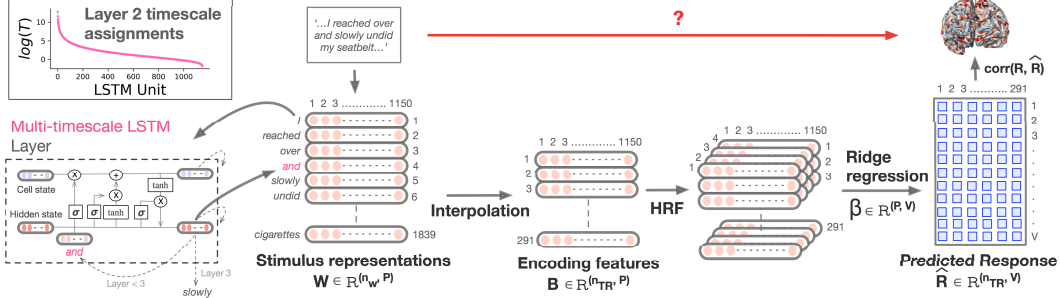

Figure 1: The multi-timescale encoding model uses hidden state representations from a modified LSTM LM to predict fMRI responses. The LSTM is first trained as an LM. Then, the hidden state is extracted for each word in the stimuli. This forms the representations $W \in \mathbb{R}^{n_w \times P}$ which are downsampled and interpolated to form the encoding model features $B \in \mathbb{R}^{n_{TR} \times P}$. The hemodynamic response function (HRF) of each voxel is approximated with a finite impulse response (FIR) model using 4 delays. Ridge regression is used to estimate the weights $\beta$ and build a predictive model for voxel response to continuous speech stimuli. Encoding model performance for each voxel is evaluated as the product-moment correlation of the predicted and true responses on a separate test stimulus. Top-left inset: the distribution of assigned timescales across units in the multi-timescale LSTM LM.

to be a *linearized* model comprising a non-linear transformation of the stimuli followed by a linear projection [15], $f(S) \coloneqq g(S)\beta$. The linearizing transform $g$ projects the stimulus $S$ into a $P$-dimensional feature space, and is selected to extract properties of the stimulus that are represented in the brain. We estimated a separate encoding model $\hat{f}_v$ for each voxel $v$ using a training dataset $(S_{train}, R_{train})$. To find the best regularization coefficient for each voxel, the regression procedure was repeated 50 times, each time holding out a random sample of 5000 TRs (125 blocks of 40 consecutive TRs) from the training set to use for validation. Ridge coefficients were selected based on validation set prediction performance, averaged across the 50 runs [4, 7]. Then we evaluated model performance with the product-moment correlation between predicted and true responses on the test dataset, $r_v = corr(\hat{R}_{v,test}, R_{v,test})$.

To test whether $r_v$ was significantly greater than 0 for each voxel $v$, we ran a block permutation test on the predicted response sequence. For permutation $k$, the predicted responses $\hat{R}_v$ were divided into blocks of 10 TRs. Then the blocks were shuffled and concatenated to yield $\hat{R}_{v,k}^{perm}$. The correlation for permutation $k$ was then computed as $r_{v,k}^{perm} = corr(\hat{R}_{v,k}^{perm}, R_v)$. This procedure was repeated for $K = 1000$ permutations, and the $p$-value for each voxel $v$ was computed as

$$p_v = \frac{1}{K} \sum_{k=1}^{K} \mathbb{I}_{r_v \leq r_{v,k}^{perm}} \tag{1}$$

where the indicator function $\mathbb{I}_{r_v \leq r_{v,k}^{perm}} = 1$ when $r_v \leq r_{v,k}^{perm}$, and 0 otherwise. To correct for multiple comparisons across all voxels within a subject, we used non-parametric False Discovery Rate (FDR) correction at $q = 0.05$ [16]. Voxels with a $p$-value lower than the FDR-corrected critical value were considered statistically significant.

## 3 Learning multi-timescale representations for language

Prior work has shown that LSTM LMs discover useful representations for many natural language processing tasks [17–21]. In particular, the encoding model transform $g(S)$ can be effectively modeled by LSTM hidden state representations [8]. We directly extend this approach by building interpretable, multi-timescale LSTM LMs for encoding models to facilitate a principled analysis of the cortical temporal hierarchy for language.

### 3.1 Multi-timescale LSTM LM

To build an explicit multi-timescale LSTM, the memory timescale of each individual unit should be controlled. The rate at which information flows into and out of memory in an LSTM is determined by the forget and input gates [10, 11]. Thus, the timescale $T_p$ of an LSTM unit $p$ can be set by fixing the forget gate bias $b_{fp}$ as $b_{fp} = -\log(e^{1/T_p} - 1)$ and the input gate bias $b_{ip}$ as $b_{ip} = -b_{fp}$ [11]. To determine the appropriate distribution of timescales we looked to a known statistical property of natural language, which is that the mutual information between tokens decays as a power law with increased token separation [1]. Although LSTM cell states exhibit exponential decay over time rather than power law [10], we can emulate a power law using an appropriate mixture of exponentials. Other work [11] shows that the correct mixture of exponentials is given by an inverse gamma distribution. Using this distribution for $T$ leads to an effective LM with interpretable and explicit representations at different timescales needed for natural language processing.

### 3.2 LM architecture and training details

The LSTM architecture is adopted from [22]. It has 3 layers, with 1150 hidden state units in the first two layers, and 400 units in the third. To ensure that the first layer only processes short timescale information, half the units are assigned $T = 3$, and half $T = 4$. Timescales in the second layer are distributed according to an inverse gamma distribution with a shape parameter $\alpha = 0.56$ and scale parameter $\beta = 1$ [11]. The forget gate biases (and hence, timescales) in the third layer are randomly initialized and trainable. This allows the network to learn transformations need to produce language modeling task outputs.

The multi-timescale LSTM LM was pre-trained on WikiText-2 [23], which has a vocabulary of 33K unique words, ∼2M tokens in the training set, 220K in the validation set, and 240K in the test set. The LM was then fine-tuned to spoken language using a separate set of stories from *The Moth Radio Hour* [13], *TED Talks*, and *Modern Love*. This dataset has 940K tokens for training, 200K for validation and 5.9K for testing. The embedding layer is re-trained during fine-tuning with a new vocabulary of 13.8K unique words that incorporates *all* words from the fMRI experiment stimuli. Further details on pre-training and fine-tuning can be found in Supplementary Section 2, including hyper-parameters that were modified from [22]. The multi-timescale LM achieved a perplexity of $68.33 \pm 0.12$ on WikiText-2 test set while the baseline LM (no timescale specification) was at $70.23 \pm 0.24$. These values are comparable to Merity et al. [21].

### 3.3 Modeling $g(S)$ with an LSTM LM

$g(S)$ is modeled by hidden state representations extracted from the LSTM LM [8] (Figure 1). Unlike previous work, we use a *stateful* LSTM that maintains the hidden and cell states between successive sequences (i.e. the context length is effectively infinite). Not only do *stateful* LSTMs perform better as language models [22], they also better represent the input stream of tokens in human language processing, providing a naturalistic setting to build encoding models.

## 4 Downsampling multi-timescale representations to BOLD responses

The fMRI encoding features for words in natural speech are modeled using a progression of three representations, each operating in a different time domain. Let $W_i$ be the word representation, where $i \in \{1, 2, \ldots, n_w\}$ is the index of the word in the stimulus and $n_w$ the total number of words. To illustrate our approach, we assume $W_i \in \mathbb{R}$. In reality, $W_i \in \mathbb{R}^P$ and the approach is applied to each of the $P$-dimensions individually. Here, $W_i$ is the LSTM hidden state activation extracted at token $w_i$ for a selected unit. Next we define $N(t)$, the putative continuous-time feature, where $t \in [0, T]$ is the elapsed time in seconds from the start of the stimulus. Our goal is to obtain $B_r$, the downsampled feature response, where $r \in \{1, 2, \ldots, n_{TR}\}$ is the corresponding volume index in the fMRI acquisition. Volumes are acquired every $T_R = 2$ seconds.

To transform $W$ to $N$, we must account for the rate of word presentation and the timescale properties of the LSTM units. To transform $N$ to $B$, we must account for the low temporal resolution of fMRI. Below, we describe the previous and proposed approaches for these transformations.

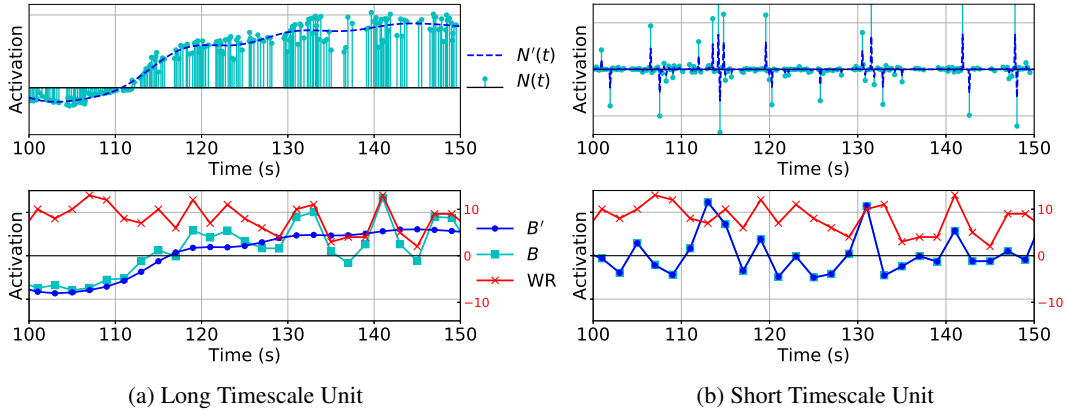

(a) Long Timescale Unit             (b) Short Timescale Unit

Figure 2: Conversion methods for long timescale (left) and short timescale (right) LSTM units. Top: in the previous method, the putative continuous-time feature value $N(t)$ is a summation of $\delta$-functions (one for each word in the stimulus) weighted by LSTM activations. The proposed interpolation of LSTM activations is $N'(t)$. Bottom: The feature activations downsampled to the rate of fMRI acquisition ($T_R = 2$ s) are $B$ from $N(t)$, and $B'$ from the interpolated $N'(t)$. We also show the word rate *WR* (red). The interpolation method captures the slow drift in long timescale units better than the previous method, while also maintaining the same quality for short timescale units. For long timescales, the previous method (cyan) is also highly correlated with word rate (red). Interpolation (blue) corrects this confound.

## 4.1 Previous method: $\delta$-functions for modeling continuous response

Let $t_i$ indicate the elapsed time in seconds from the start of the story when $w_i$ was spoken. As in previous work [4, 7], $N(t)$ is defined as a sum of $\delta$-functions at the time of each word weighted by the corresponding word representation, $N(t) = \sum_{i=1}^{n_w} W_i\, \delta(t - t_i)$. To transform $N$ to $B$, we first apply an anti-aliasing, low-pass Lanczos filter $L(t)$ to $N(t)$ in continuous time,

$$N_{LP}(t) = N(t) * L(t) = (N * L)(t) = \int_{-\infty}^{\infty} N(\tau)L(t - \tau)d\tau, \qquad (2)$$

and then sample $N_{LP}$ at times $t_r$ where $r \in \{1, 2, \ldots, n_{TR}\}$ to give $B$, i.e., $B_r = N_{LP}(t_r)$.

**Direct transformation of *W* to *B*.** Since $N(t)$ is modeled as a sum of delta functions, the steps above can be simplified by defining a matrix $\mathbf{L}$ such that $\mathbf{L}_{ir} = L(t_i - t_r)$. $B$ can then be computed with a matrix multiplication as $B = W\mathbf{L}$ (Supplementary Section 2).

**Issues with this approach.** We refer to the above as the $\delta$-sum technique. While this is a common encoding model approach, it operates under the assumption that the underlying response to a word $w_i$ is an infinitesimal spike at $t_i$. Consequently, it assumes that $N(t)$ can be reduced to a discrete summation. These assumptions hold when $W$ is rapidly changing, as is the case for a short-timescale unit (Figure 2(b)). However, for long-timescale units, $W$ drifts slowly over time, and these assumptions become invalid. Moreover, these assumptions lead to $N(t)$ being highly confounded with the spoken word rate, which is particularly undesirable for long-timescale representations (Figure 2(a)). Instead, we would like $N(t)$ to vary smoothly across time for long-timescale representations.

## 4.2 Gaussian RBF kernels for modeling long-timescale representations

We can solve these issues by generalizing the previous $\delta$-sum method, which can be viewed as a specific instance of *interpolation*. Here we use a kernel function $k_\epsilon(t)$ to create a new putative continuous time function $N'(t)$ that interpolates the values in $W$,

$$N'(t) = \sum_{i=1}^{n_w} a_i k_\epsilon(t - t_i). \qquad (3)$$

Note that the original word representations $W_i$ are replaced by new interpolation weights $a_i$. To find these weights, we first define the matrix $\boldsymbol{\Phi}$ where $\boldsymbol{\Phi}_{ij} = k_\epsilon(t_i - t_j)$. Then, the corresponding linear system is solved to give $a = \boldsymbol{\Phi}^{-1}W$. We define the vector $\mathbf{k}_i(t) = k_\epsilon(t - t_i)$ so $N'(t)$ can be written as the matrix product

$$N'(t) = \mathbf{k}(t)\boldsymbol{\Phi}^{-1}W. \tag{4}$$

We use a Gaussian radial basis function (RBF) kernel $k_\epsilon(t) = e^{-(\epsilon t)^2}$ with shape parameter $\epsilon > 0$. Note that the original $\delta$-sum method is a special case of interpolation: if $\epsilon \to \infty$, then $\boldsymbol{\Phi} \to I$, so $N'(t) \approx \mathbf{k}(t)W = N(t)$.

To obtain the downsampled feature vector $B'$ from $N'(t)$ we follow the previous approach (Equation 2; Supplementary Section 2), giving

$$B'_r = (N' * L)(t_r) = \int_{-\infty}^{\infty} \left[ \sum_{i=1}^{n_w} a_i \, k_\epsilon(\tau - t_i) \right] L(t_r - \tau)d\tau. \tag{5}$$

**Generalized direct transform of $W$ to $B'$.** Earlier, the matrix $\mathbf{L}$ was computed as the value of the Lanczos filter for each combination of word time $t_i$ and fMRI sample time $t_r$. Since $N'(t)$ is no longer a sum of $\delta$-functions, we need to consider every time point, not just the word times $t_i$.

Consider a set of $n_f$ very finely-spaced timepoints, $t_f$. We define a new Lanczos matrix $\mathbf{L}_f \in \mathbb{R}^{n_f \times n_{TR}}$ and a kernel matrix $\mathbf{k}_f \in \mathbb{R}^{n_w \times n_f}$ where the $i, f$ entry is given by evaluating the kernel function for the corresponding word time $t_i$ and finely-spaced timepoint $t_f$. Then, the convolution integral can be approximated as $\mathbf{k}_f \mathbf{L}_f$ and $B'$ can be obtained using matrix multiplication,

$$B' = W^\top \boldsymbol{\Phi}^{-1} \mathbf{k}_f \mathbf{L}_f. \tag{6}$$

**Free parameters.** The kernel shape parameter $\epsilon$, given in seconds$^{-1}$, determines the time duration over which responses are integrated. It should thus be related to the timescale of the LSTM unit which generates the feature representation $W_i$. We select $\epsilon = \frac{1}{T_p d}$, where $T_p$ is the timescale assigned to the unit (in words) and $d$ is the average seconds per word across the experimental stimuli. This ensures that the kernel width is linked to each unit's timescale by the average rate of the natural speech data itself. The interpolation weights $a_i$ are found by solving for $a$ in $a = \phi^{-1}W$ using ridge regression. The ridge coefficient is estimated by leaving one word out at a time and measuring accuracy of interpolating its activation from other words. Interpolation is done separately for each LSTM unit.

In practice, many kernels could be used for interpolation. However, the RBF kernel 1) generalizes to the $\delta$-sum method and 2) has a kernel width that can be directly linked to timescale. These properties are not exhibited by other commonly used alternatives, like polyharmonic spline kernels.

## 5 Using interpretable LSTMs to estimate voxel timescales

Recall that the encoding model is given by $f(S) \coloneqq g(S)\beta$, where $\beta$ is a set of $P$ linear weights. In this work, $g(S)$ is modeled by the LSTM hidden state at each word in the stimulus. Since $f$ is a linear transformation of $g$, the norm of $\beta$ values represent the *feature importance* of the associated hidden unit. Since each unit in LSTM layer 2 is assigned a fixed timescale, a large weight value in $\beta$ thus indicates that the timescale of that unit and consequently, the information it encodes, are relevant for the given voxel. The relative feature importance across different timescale units can then reliably estimate the voxel's timescale.

The estimated timescale for voxel $v$ is $T_v$,

$$T_v = e^{\frac{\sum_{p=1}^{P} \beta_p^2 log(T_p)}{\sum_{p=1}^{P} \beta_p^2}} \tag{7}$$

where $\beta_p$ is the weight on unit $p$ and $T_p$ is its assigned timescale. Thereby, the voxel timescale $T_v$ can be interpreted as the average number of words that a voxel integrates over in an input sequence.

In supplementary section 1.3, $|\beta|$ is visualized for two example voxels estimated to have short and long timescales respectively. For the short timescale voxel, the high magnitude weights are concentrated on LSTM units with shorter assigned timescales, while the opposite is true for the long timescale voxel. This provides the basis for quantifying voxel timescales based on the effective feature importance ($\beta$) of LSTM units at different timescales.

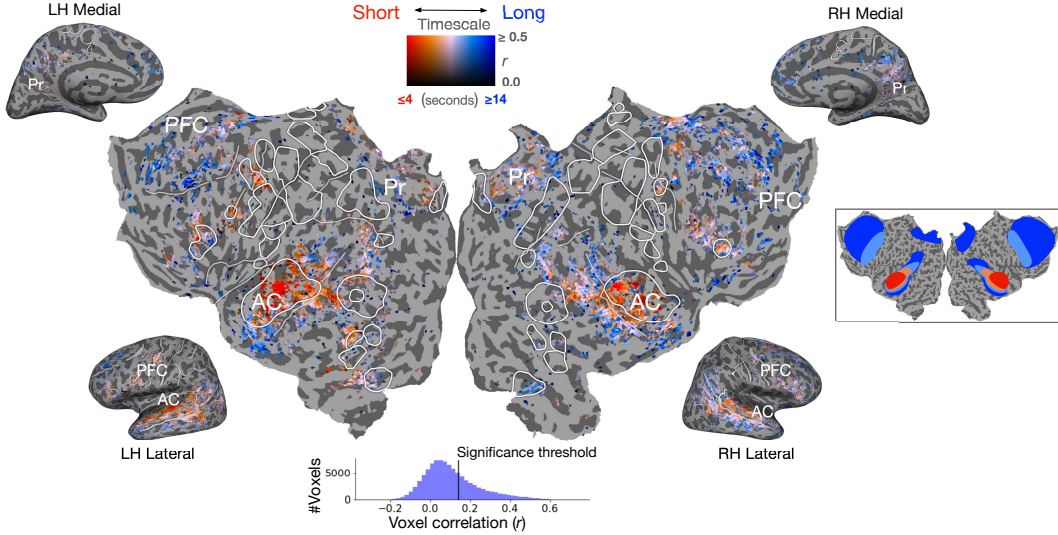

Figure 3: Estimated timescale $T_v$ across cortex for significant voxels ($p < 0.05$, FDR-corrected) in the MT model. Voxels with longer estimated timescales are shown in blue, and those with short timescales are shown in red on the flatmap. Non-significantly predicted voxels are gray. The right inset shows a schematic of timescale estimates from previous work [2] based on stimulus scrambling. Our approach corroborates patterns highlighted in the inset, but provides a more detailed account of timescales across the cortex. The bottom inset shows a histogram of voxel correlations and the significance threshold. AC: Auditory Cortex, Pr: Precuneus, PFC: Prefrontal cortex. Similar maps for other subjects are shown in Supplementary Section 1.

## 6   Results

The multi-timescale (MT) encoding model uses a stateful, multi-timescale LSTM followed by RBF interpolation to model $g(S)$. Across cortex, this model explains similar amounts of variance in the fMRI data as other modeling variants. However, the advantage of this approach lies in the interpretability it offers. All additional results can be found in Supplementary Section 1.

### 6.1   Using explicit timescale information in LSTM units to infer temporal selectivity

We define $T_v$ to be the estimated timescale of a voxel $v$. Previous work that uses scrambling experiments to estimate timescales can only make coarse distinctions in $T_v$ based on the response of a voxel (or lack thereof) to stimuli that are scrambled at different temporal scales [2]. However, multi-timescale language encoding models represent stimuli in a high-dimensional, densely sampled temporal feature space. This facilitates a principled and fine-grained estimation of $T_v$ as described in Section 5. Figure 3 shows $T_v$ values for significantly predicted voxels (bottom inset) compared to a schematic of earlier results [2, 24]. Our $T_v$ estimates vary smoothly across the temporoparietal axis ranging from short timescales in the auditory cortex (AC) to longer timescales in the inferior parietal region. Further, the prefrontal (PFC) cortex has increasingly long $T_v$ from the central sulcus (CS) to more anterior regions. In the precuneus (Pr), we observe a medium to long timescale gradient along a ventral to dorsal axis. These patterns are broadly in agreement with prior findings (inset, Figure 3) [2], but offer additional fine-grained detail.

### 6.2   Interpolating encoding model features improves timescale estimates

The motivation behind the interpolation technique described in Section 4 is to accurately represent long timescale features in fMRI encoding models. To illustrate the usefulness of this technique, we compared $T_v$ estimates from the MT model (with RBF interpolation) to its $\delta$-sum counterpart. The $\delta$-sum encoding model also used the stateful, multi-timescale LSTM to model $g(S)$. However, similar to previous encoding approaches [4, 7, 8], it downsampled stimuli representations by summing across $\delta$-functions. The estimated timescales are compared in Figure 4 for selected regions of interest (ROIs,

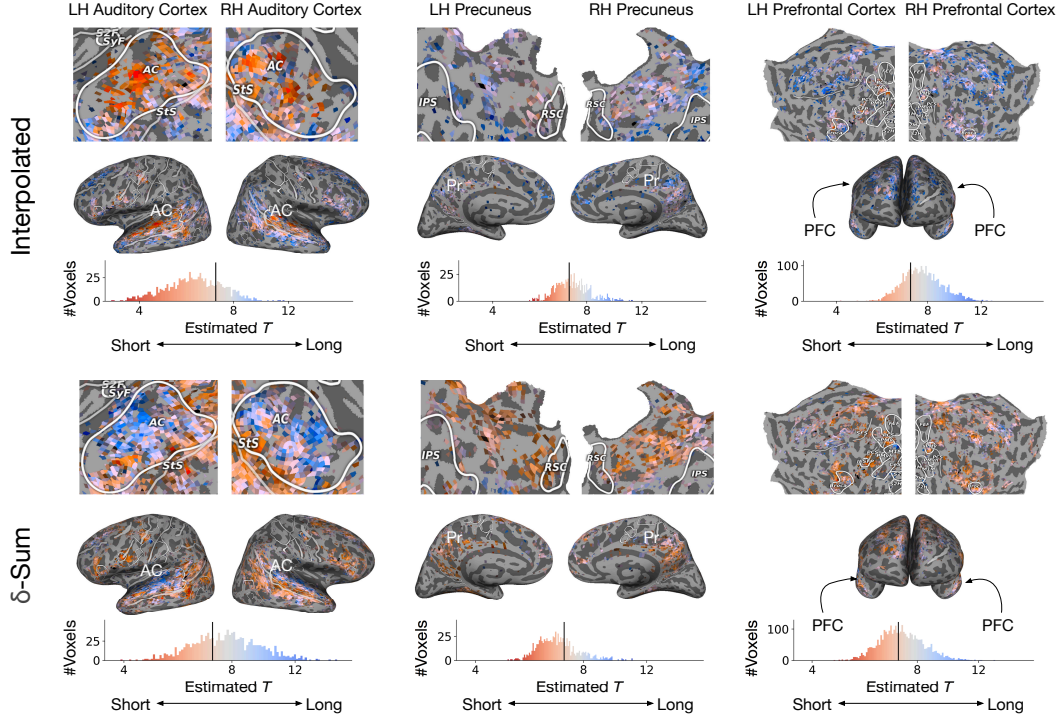

Figure 4: Comparing estimated timescales $T_v$ across two different temporal downsampling schemes. Both use stateful, multi-timescale LSTMs (layer 2) to model $g(S)$. One scheme downsamples representations by interpolating with an RBF kernel while the $\delta$-sum model sums across $\delta$-functions. Histograms show the distributions of $T_v$ for significant voxels in each ROI across all 6 subjects. The black vertical line in the histogram shows the mean $T_v$ across cortex for each downsampling scheme. The flatmap and inflated 3D brain are for a single subject. In the AC, $\delta$-sum inaccurately assigns more medium to long timescales. In the precuneus and PFC, the $\delta$-sum model overrepresents short timescales. This highlights the drawbacks of the $\delta$-sum approach for creating encoding features that operate at different timescales. In contrast, the RBF interpolated model appropriately estimates timescales in all brain regions. Colormap follows Figure 3. Similar maps for other subjects are shown in Supplementary Section 1.

defined by procedure in Supplementary Section 3). Multiple lines of evidence demonstrate that primary AC in humans processes short timescales during natural language processing [2, 3, 25, 26]. Contrary to this, the $\delta$-sum estimates in primary AC are biased towards medium and long timescales. This is likely because long-timescale representations are strongly correlated with word rate when downsampled using the $\delta$-sum method (Figure 2). Further, the MT model estimates longer timescales in higher-order regions like precuneus and PFC (Figure 4). This is in line with both experimental evidence from language tasks [2, 3, 25] and hierarchical timescale theory based on evidence from intrinsic timescale estimates and network connectivity [27–30].

## 6.3 Direct voxel timescale estimates improve upon indirect context-length manipulations

Previous work used an indirect context length (CL) manipulation to estimate $T_v$ [8]. To replicate this scheme using our model, representations were extracted from a stateless LSTM (no timescale assignments) at different context lengths. Separate encoding models were built for each CL with $\delta$-sum downsampling. Prior work restricted the maximum context length to 19 words. To test for longer timescales, we used context lengths [0, 2, 4, 8, 16, 32, 64].

As described in [8], a voxel's CL preference is estimated as the center of mass of the encoding performance curve across different CLs. Figure 5 shows the voxel timescales estimated by CL manipulation. Note the x-axis on the histogram is for CLs 0 to 64 and hence, is not directly equivalent to previous figures. As discussed in the supplement, if the performance across CLs is similar (curve

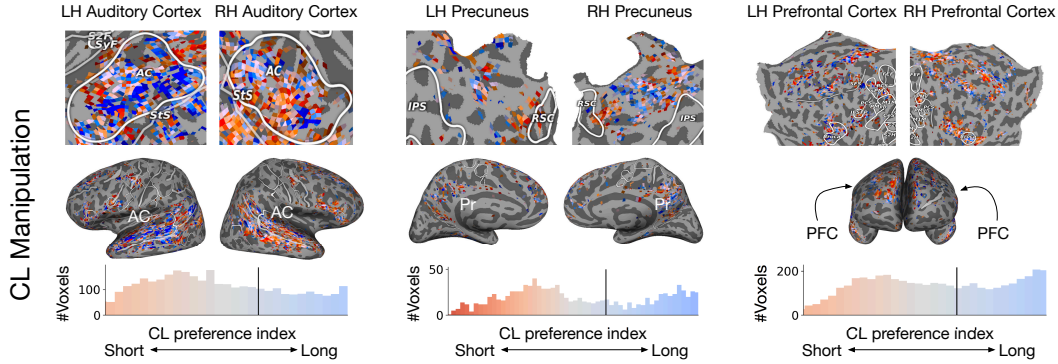

Figure 5: Estimating timescale by manipulating the context length (CL) is a less interpretable method. A stateless LSTM was used to create encoding models for CLs [0, 2, 4, 8, 16, 32, 64]. The timescale of each voxel was estimated with a CL preference index [8]. Only voxels significant in *all* CL models are shown. In AC, some voxels have long CL preferences. Further analysis reveals that long CL representations still retain short-timescale information. Similar maps for other subjects are shown in Supplementary Section 1.

is flat, Supp. Fig 1A), the voxel has a large center-of-mass, artificially inflating the CL preference. This is the case with many AC voxels (Fig. 5) that are shown to have a preference for *long* CLs. It also suggests that even long CL representations retain relevant short-timescale information that reliably predicts voxel responses. This illustrates that manipulating the CL does not change the timescale of the representation as a whole. While it appropriately assigns mid-range CL preferences for PFC and the precuneus, the CL estimates are still difficult to map directly to the timescale $T_v$ of a voxel. In sum, the CL method can provide evidence for a temporal hierarchy in the cortex, as shown previously [8]. However, we show that shortcomings of this method are addressed by the MT model that yields more accurate and interpretable estimates of voxel timescale.

# 7 Conclusion

This work presents a multi-timescale encoding model for predicting fMRI responses to natural speech. To create the encoding model features, representations are extracted from a modified LSTM LM with explicitly fixed timescales for each hidden unit. This facilitates direct estimation of the timescale of information represented in voxel based on the encoding model weights, revealing a fine-grained map of timescale selectivity across human cortex. We show that the new method assigns timescales more accurately than previous methods that relied on changing the length of the context to the LSTM LM. The fine-grained map also provides basis for further investigation of language-specific timescale gradients in regions such as the precuneus. Our work creates a framework for developing interpretable, LM-based encoding models that can be used to formulate and test new hypotheses about timescale representations in the brain. For example, future investigations could examine how perturbing inputs to the LM affects encoding performance. Lastly, we modified the standard encoding model approach for mapping between time domains in order to accurately capture LSTM representations at different timescales. Our new interpolation method improved timescale estimates across a variety of brain regions. This can be generalized for use in any encoding model study investigating timescale representations in the brain, and is not specific to natural language or LSTMs. Our work illustrates that improving the interpretability of neural networks can yield more interpretable formal models of the brain.

## Broader Impact

Researchers working to understand temporal phenomena may consider the problems raised in this work, and may find the proposed methods useful for their analysis. More importantly, this work is a stepping stone towards building better models for language processing in the brain that can not only help investigate cortical language processing but also simulate brain responses. This could be useful for diagnosing, treating, and assisting people with language deficits like aphasia, especially since

processing information at different timescales is critical to human language. On the contrary, these tools may also serve as a stepping stone toward unethical brain decoding practices that could be used by, for example, insurance companies or attorneys for erroneous evidence collection on a trial. In general, advances in brain-reading technology may raise issues in neuroethics, especially regarding mental privacy.

Negative consequences from this research may affect the participants themselves. The fMRI data for this work was acquired in accordance with IRB protocols, which included informed consent of the risks involved with MRI. In addition to physical risks, such as peripheral nerve stimulation, participants were informed about the steps taken to protect their data. While personal identifying information about participants is stored in a physical, locked, separate location from the neuroimaging data, a failure in this system could potentially lead to a breach of confidentiality.

As with much of the research submitted to NeurIPS, training neural network models consumes large amounts of energy. If this energy was generated by non-renewable fuel, this would have a negative impact on the environment.

## Acknowledgments and Disclosure of Funding

We would like to thank Nicole Beckage and RJ Antonello for valuable feedback on the manuscript and useful discussions, Lauren Wagner for annotating the experimental stimuli; and the anonymous reviewers for their insights and suggestions. Funding in direct support of this work: the Burroughs-Wellcome Fund Career Award at the Scientific Interface (CASI), Intel Research Award, HPC resources provided by the Texas Advanced Computing Center (TACC) at The University of Texas at Austin, Alfred P. Sloan Foundation Research Fellowship, and GPUs donated by NVIDIA. Alexander G. Huth also holds a position at Caseforge, Inc., whose products were used in the fMRI experiment.

## Footnotes

*Currently: Helen Wills Neuroscience Institute, UC Berkeley

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
