[Supplementary Material]

# Supplementary Material: Interpretable multi-timescale models for predicting fMRI responses to continuous natural speech

**Shailee Jain**
Department of Computer Science
The University of Texas at Austin
Austin, TX 78712
shailee@cs.utexas.edu

**Vy A. Vo**
Brain-Inspired Computing Lab
Intel Labs
Hillsboro, OR 97124
vy.vo@intel.com

**Shivangi Mahto**
Department of Computer Science
The University of Texas at Austin
Austin, TX 78712
shivangi@utexas.edu

**Amanda LeBel** [*]
Department of Neuroscience
The University of Texas at Austin
Austin, TX 78712
amanda.lebel@berkeley.edu

**Javier S. Turek**
Brain-Inspired Computing Lab
Intel Labs
Hillsboro, OR 97124
javier.turek@intel.com

**Alexander G. Huth**
Computer Science & Neuroscience
The University of Texas at Austin
Austin, TX 78712
huth@cs.utexas.edu

## 1 Auxiliary results and analysis

### 1.1 Individual subject flatmaps

Additional subject flatmaps are shown in figures 2-7 at the end of the document. Each of these figures has 3 flatmaps for a subject that depict the estimated timescale in the interpolated and $\delta$-sum models, and the CL preference index as per the CL manipulation technique. Only significantly predicted voxels are shown. These flatmaps correspond to figures 3-5 in the main text and follow the same colormap. Note that subject S04 is excluded from this study due to poor data quality, resulting in 6 subjects overall. Results from subject S03 (highest number of significant voxels) are shown in the main text. These additional flatmaps corroborate the claim that the interpolated model is able to effectively reveal the temporal hierarchy across the cortex for language processing. Further, both the $\delta$-sum and CL manipulation approaches make erroneous estimates in several parts of the cortex, highlighting the importance of interpretable multi-timescale representations and the RBF interpolation scheme.

### 1.2 Plateau in $r$ for AC voxels with long CL preference

Section 6.3 in the main text discusses the CL manipulation technique in *stateless* models to investigate the cortical temporal hierarchy. It is noted that several voxels in the auditory cortex (AC) have a preference for *long* CLs. This pattern is confirmed in other subjects as seen in the 'CL Manipulation' flatmaps in figures 2-7. To investigate this, we look at the encoding performance of different CL models. For each voxel, this is given by the linear correlation $r$ of the predicted and true responses.

---

[*]Currently: Helen Wills Neuroscience Institute, UC Berkeley

Figure 1: (A) Encoding model performance $r$ for some voxels in the auditory cortex that have *long* CL preference. The performance plateaus after CL 4 suggesting that long CL representations retain relevant short-timescale information. This highlights that CL manipulation is an indirect way of controlling the timescale of a representation and can often lead to erroneous estimations. (B) Regression weights across LSTM units for two voxels estimated to have *short* and *long* timescales respectively. For the *short* (*long*) timescale voxel, the LSTM units with shorter (longer) timescales have high weight norms. This forms the basis of our estimation procedure that uses the regression weight and assigned timescale of each LSTM unit to compute the timescale of a voxel.

Figure 1(A) shows a few voxels in the auditory cortex that have *long* CL preference. It is clear that for all voxels the encoding performance plateaus after CL 4 and the *long* CL preference is a caveat of the estimation procedure. The plateau also suggests that even long CL representations retain relevant short-timescale information to reliably predict voxel responses. Thus, manipulating the context length does not change the timescale of the representation as a whole. This limits the usefulness of the CL manipulation technique.

## 1.3 Representative weight profiles for *short* and *long* timescale voxels

In section 5 of the main text, we introduce a timescale estimation procedure based on the regression weight $\beta_p$ and the assigned timescale $T_p$ for each LSTM unit $p$. Figure 1(B) shows $|\beta|$ for two example voxels estimated to have short and long timescales respectively. For the short (long) timescale voxel, the high magnitude weights are concentrated at LSTM units with shorter (longer) assigned timescales. This provides the basis for quantifying voxel timescales based on the effective feature importance ($\beta$) of LSTM units at different timescales.

## 1.4 Encoding model performance

The overall encoding performance of a model is given by the average $r$ across voxels in all subjects and error is given by the Standard Error of Mean (SEM). The MT model uses a stateful LSTM to model $g(S)$ followed by RBF interpolation and has a performance of $0.068 \pm 0.0043$. The $\delta$-sum model uses the same $g(S)$ but down-sampling is done by the $\delta$-sum method. This model has a performance of $0.035 \pm 0.0039$. For the CL manipulation experiment, a stateless LSTM is used to model $g(S)$ at different context lengths followed by $\delta$-sum down-sampling. Table 1 shows the encoding performance for different CL models. These results suggest that the various encoding model do not differ too much in performance. However, our main analyses show that unlike other models and methods, the MT model specifically controls timescales of LSTM units and effectively represents long-timescale information. The main contribution of this paper is the added interpretability and effective handling of long-timescale information that facilitate investigation of voxel timescales to a finer and more reliable degree.

Table 1: Average $r$ across voxels in all subjects with standard error of mean (SEM) for different CL models. Each model uses representations extracted from a stateless LSTM (no timescale assignment) at the specified context length, followed by $\delta$-sum down-sampling.

| CL 0 | Cl 2 | CL 4 | CL 8 | CL 16 | CL 32 | CL 64 |
|---|---|---|---|---|---|---|
| 0.029±0.0011 | 0.027±0.0006 | 0.028±0.0006 | 0.027±0.0004 | 0.025±0.0003 | 0.024±0.0008 | 0.021±0.0018 |

# 2 Multi-timescale encoding models

## 2.1 Training and fine-tuning multi-timescale LSTM

**Hardware specifications and run time** The LSTM language models were trained using a single GeForce GTX TITAN X GPU with 64GB CPU RAM. All code was written in pytorch

**LSTM hyper-parameters** The network is trained with SGD followed by non-monotonically triggered average SGD

**Fine-tuning datasets** Training set: The Moth Radio Hour [2], *TED Talks*[3], and *Modern Love*[4]; Validation set: Train split of fMRI stimuli; Test set: Test split of fMRI stimuli.

## 2.2 Feature interpolation

**Details on direct transform of** $W$ **to** $B$**.** An efficient solution for this set of transformations skips explicit evaluation of these different representations by noting that convolving with $\delta$-functions "sifts" out the corresponding values,

$$B_r = (N * L)(t_r) \tag{1}$$

$$= \int_{-\infty}^{\infty} \left[ \sum_{i=1}^{n_w} W_i \, \delta(\tau - t_i) \right] L(t_r - \tau) d\tau \tag{2}$$

$$= \sum_{i=1}^{n_w} W_i \int_{-\infty}^{\infty} \delta(\tau - t_i) L(t_r - \tau) d\tau \tag{3}$$

$$= \sum_{i=1}^{n_w} W_i \, L(t_i - t_r) \tag{4}$$

Thus if we define the matrix $\mathbf{L}$ where $\mathbf{L}_{ir} = L(t_i - t_r)$, we can compute $B$ with a single matrix multiplication,

$$B = W\mathbf{L}. \tag{5}$$

# 3 MRI acquistion, preprocessing, and experiment details

## 3.1 Participants

All participants were healthy and had normal hearing, and normal or corrected-to-normal vision. To stabilize head motion during scanning sessions participants wore a personalized head case that precisely fit the shape of each participant's head (`https://caseforge.co/`). Anatomical data for subject S-02 were collected on a 3T Siemens TIM Trio scanner using a 32-channel Siemens volume coil at a different site. The same MP-RAGE sequence was used.

## 3.2 Stimulus preparation and presentation

Story stimuli were played over Sensimetrics S14 in-ear piezoelectric headphones. The audio for each story was filtered to correct for frequency response and phase errors induced by the headphones using calibration data provided by sensimetrics and custom python code [5] All stimuli were played at 44.1 kHz using the pygame library in Python (https://www.pygame.org/news).

## 3.3 Acquisition parameters

Whole-brain MRI data was collected on a 3T Siemens Skyra scanner using a 64-channel Siemens volume coil. Functional MRI (fMRI) data were collected using a gradient echo EPI sequence, multi-band factor of 2. Scan parameters included repetition time (TR)=2.00s, echo time (TE)=30.8 ms, flip angle=71$°$, voxel size=2.6mm$^3$, matrix size=84x84, field of view=220 mm. Anatomical MRI data were collected with a T1-weighted multi-echo MP-RAGE sequence with voxel size=1mm$^3$.

## 3.4 Processing MRI data

**Functional data.** All functional data were motion corrected using the FMRIB Linear Image Registration Tool (FLIRT) from FSL 5.0. FLIRT was used to align all data to a template that was made from the average across the first functional run in the first story session for each subject. These automatic alignments were manually checked for accuracy.

Low frequency voxel response drift was identified using a 2nd order Savitzky-Golay filter with a 120 second window and then subtracted from the signal. To avoid onset artifacts and poor detrending performance near each end of the scan, responses were trimmed by removing 20 seconds (10 volumes) at the beginning and end of each scan, which removed the 10-second silent period and the first and last 10 seconds of each story. The mean response for each voxel was subtracted and the remaining response was scaled to have unit variance.

**Anatomical data.** Cortical surface meshes were generated from the T1-weighted anatomical scans using FreeSurfer

The auditory cortex ROI was defined using a functional localizer task. Auditory cortex localizer data were collected in one 10 minute scan. The subject listened to 10 repeats of a 1-minute auditory stimulus each containing 20 seconds of music (Arcade Fire), speech (Ira Glass, This American Life), and natural sound (a babbling brook). Voxels that responded reliably to any auditory stimulus (by an F-statistic over 10 repeats) were labeled as part of auditory cortex.

The precuneus and prefrontal cortex (PFC) ROIs were defined by the Freesurfer parcellation for each subject. The PFC ROI was a combination of the superior frontal, rostral middle frontal, frontal pole, and caudal middle frontal parcellations.

Figure 2: Subject S01. Colormap follows Figure 3 in main text.

**S02**

Interpolated

δ-Sum

CL Manipulation

Figure 3: Subject S02. Colormap follows Figure 3 in main text.

**S03**

Figure 4: Subject S03. Colormap follows Figure 3 in main text.

**S05**

Interpolated

δ-Sum

CL Manipulation

Figure 5: Subject S05. Colormap follows Figure 3 in main text.

Figure 6: Subject S06. Colormap follows Figure 3 in main text.

Figure 7: Subject S07. Colormap follows Figure 3 in main text.

## Footnotes

[2]https://themoth.org

[3]https://www.ted.com/talks

[4]https://www.npr.org/podcasts/469516571/modern-love

[5]Anonymized link.