[Reviews · NeurIPS 2020]

Review 1

Summary and Contributions: This work presents a multi-timescale encoding model for predicting fMRI responses to natural speech. To create the encoding model features, representations are extracted from a modified LSTM LM with explicitly fixed timescales for each hidden unit. This makes it possible to estimate the timescale of information represented in voxel based on the encoding model weights, which creates a brain maps of timescale selectivity across human cortex. The face validity of the time scales estimates is assessed on a dataset of 6 subjects.

Strengths: The paper is well-focused on a particular question related to language processing in the human brain. It makes a clever use of current NLP technology, namely some ongoing developments on LSTM models. The paper stresses the importance of interpolation scheme in deep NLP encoding models, which I find interesting. The paper is relatively clear and well-illustrated. I enjoyed reading it.

Weaknesses: The proposed estimator of time scale (6) seems problematic, given that some beta coefficients can be negative, making the whole estimate wrong or misleading. I don't find it realistic that some regions integrate speech over 14s ! To me this result sounds more like an artifact of the estimator. It is never explained clearly why the estimator should be better than previous alternatives. I don't think it is. Experiments rests on a very peculiar dataset, with only 6 subjects. The authors avoid to use modern NLP models, such as BERT or GPT2. It is known that these achieve much higher accuracy than LSTM (e.g. https://www.biorxiv.org/content/10.1101/2020.06.26.174482v1). I agree that these may not allow to draw meaningful conclusion on integration scales, but the fact that there exist much more effective language representations than LSTM mitigates the conclusions of the present paper. Training is done in small corpora, biased toward the content used in the fMRI experiments. It should be investigated whether other, more standard ways of training LSTMs give similar results.

Correctness: Some claims are not well backed up by observations 1. There seems to be little consistency of the time scale maps across individuals, which is completely ignored in the main text (results are only shown in supplementary materials) 2. Method comparison is completely qualitative, to the point where it becomes almost arbitrary. Given that I find the conclusion mistakenly assertive. The permutation test used to assess statistical significance is flawed: the permutation scheme disrupts the temporal structure of the data, leading to underfit. This means that the non-permuted will always have a better fit, even if it explains nothing in the data. Besides FDR cannot be computed accurately from only 1000 permutations Measuring model accuracy by performance among significant voxels sounds circular; a better way should be found.

Clarity: Overall good. Given the explanation provided, I could not really make sense of the following "The weights a i are found using ridge regression, where the optimal shrinkage parameter is chosen by leave-one-out cross-validation across the n w words." this seems to play an important role and this is unclear.

Relation to Prior Work: The paper seems to build mostly from [11], adding a biological validation to the model designed there. I think that's fine. What the paper does not acknowledge well enough is the current success of Transformers in fitting the human language system.

Reproducibility: Yes

Additional Feedback: Given the complexity of the whole procedure, I think that a detailed recapitulation of the analysis procedure is needed. Worse, the data are not publicly available. 27 stories (26 train, 1 test): if the authors mean that there was a unique test set, and that no cross-validation was performed, I consider that this essentially invalidates all inferences made on this dataset. You cannot base any serious conclusion on such a narrow setup. "the δ-sum estimates in primary AC are biased towards medium and long timescales. This is likely because long-timescale representations are strongly correlated with word rate when downsampled using the δ-sum method (Figure 2). In the precuneus, the ‘δ-sum’ model estimates more short-timescale voxels. In other regions, the MT model estimates more long timescales." I think that there is no ground truth for these quantities, especially not from the literature; It is impossible to conclude that an approach is superior to another.


Review 2

Summary and Contributions: I have read the author response, read the other reviews, participated in the discussion with the reviewers and area chair, and adjusted my score accordingly. The authors tackled the problem of estimating the time scales at which various brain locations are responding to spoken speech stimuli (either directly or through integration over time or of internal processing). They based their approach on LSTM-based language models where the units in their internal representation can be ascribed a specific time scale. These were then used as the mapping from stimulus to representation in an encoding model of fMRI data. Notably, this means that the model can both predict a voxel and also provide a precise estimate of its temporal timescale. The authors show that this yields results consistent with neuroscience expectations, and with previous work estimating such timescales in a coarser manner.

Strengths: strengths: - very clear paper (with the caveat that I've read the preceding work by the authors, and almost all the related work) - first demonstration that time scale can be directly estimated, rather than by manipulating stimuli or manipulating the amount of context available to the language model

Weaknesses: weaknesses: - some of the comparisons with past approaches are elaborated on to a degree that is not necessary to understand what was done; it would be better to have more detail on the more recent approach, if someone wants to reproduce it - the discussion does not really separate clearly what is suggestive from what is ground truth (to the extent known)

Correctness: Yes. The sole omission is the test for significance of how well a voxel can be predicted, which is in the appendix. It's correct, but should be in the main text.

Clarity: Generally, yes. Specific questions: 111: What happens to the rest of the vocabulary? How critical is this particular step? 117: By "consecutive sequences", do you mean successive sentences? If so, is the state maintained as a single accumulator vector within the LSTM, or in a different way? What are the implications of "infinite context length" for brain activation, given that your stimuli last many minutes? 4) I suspect that readers not familiar with the approach described in 4.1 (which is used in previous papers sharing at least one author...) will not understand the point of 4.2, given the sparsity in detail. It might make sense to sacrifice part of Figure 2 to illustrate the basic principle, rather to compare it with the new method. 6.1) In Figure 3 you show the estimated timescale for voxels, and mention they are significant. Presumably this means that, in those voxels, the prediction correlated with the true time series to a degree that would not happen by chance. But what are the correlations? Where is information about the test being performed? Since you use this to select all the voxels shown, it should be in the main text. 6.2) In Figure 4 there are a number of statements to the effect of \delta-sum being inaccurate or overrepresenting timescales. In the text, 215 says "contrary to literature". Are the findings in [2,3,24] being treated as ground truth and, if so, how reasonable is this? 225: How was context length restricted?

Relation to Prior Work: Yes, the main papers looking at using stimulus or language model manipulations to indirectly infer the degree to which a voxel responds to a timescale are discussed. This is primarily about encoding rather than decoding models, so those suffice. The LSTM with explicit timescale for each unit is introduced and, to my knowledge, that's the only such model.

Reproducibility: Yes

Additional Feedback: It is pretty obvious that the authors are in the Huth group, be it because of the methodology, the dataset, or the figures being very similar to a preceding paper. That said, they did not go out of their way to blow anonymity.


Review 3

Summary and Contributions: Update after rebuttal: In my original review, I asked for a better motivation for the voxel timescale estimate provided by the authors in Eq. 6. R1 also appears to have concerns about this timescale estimation. As it turns out, there was a typo in Eq. 6 (pointed out in the rebuttal). However, the authors do not offer more motivation about using this specific estimate of the voxel timescale. I am glad they provided a visualization in Fig. 1B in the supplementary (which should be referenced in the main paper) that is similar to what I suggested but it’s not clear to me that the visualization actually shows what the authors conclude. For example, it’s not clear whether the two plots have the same scale of beta magnitudes. Also it seems that the vast majority of LSTM units have short timescales, in both the long- and short-timescale-assigned voxels. This may be one reason why there is virtually no voxels that have long timescales in 4 out of the 6 subjects in the new plots provided in the rebuttal (after a bug fix). Overall my confidence in this paper is a bit shaken by the combination of: 1. the presence of at least one bug and several important typos in the main text, 2. the extreme complexity of the procedure (as R1 also points out), and 3. the fact that the method heavily relies on a non-peer-reviewed method that the authors have concurrently submitted. However, I am willing to give this work the benefit of the doubt in the hope that future research will validate and replicate the authors' results, so I maintain my score. *************************** This paper presents an advancement in encoding models for predicting fMRI recordings elicited by natural language stimuli. Specifically, this work develops a new approach to combine word representations from an LSTM in a way that honors the representations' timescale, in an attempt to reveal the timescales of language processing of individual fMRI voxels. To obtain LSTM representations with different timescales, the authors make use of a previously developed approach that modifies LSTM gates such that individual hidden units can encode a different timescale of language information.

Strengths: The proposed approach is a significant advancement in encoding models for fMRI recordings of natural language processing stimuli, as it allows for a more precise manipulation of the timescale encoded in the stimulus representation. The authors present a comprehensive empirical evaluation of their proposed approach against alternative ways of combining word representations, and previous work

Weaknesses: This work is relevant to a small fraction of the NeurIPS community which works with encoding models for fMRI, specifically in language processing There are a few key places where the clarity can be improved, especially in Section 4.2 and Section 5. It would be helpful if the authors give more context for why they made certain computational choices. For example, why was the RBF kernel chosen as opposed to a different kernel?

Correctness: Yes

Clarity: Mostly, see under weaknesses.

Relation to Prior Work: Yes

Reproducibility: Yes

Additional Feedback: It's important that the altered multi-timescale LSTM can encode the actual language statistics well. Can the authors say how well the LSTM actually performs at the task it was trained to do? For example, what is the perplexity of the pretrained model on the wiki testset and of the fine-tuned model on the spoken language stimuli? Fig 1 mentions that ridge regression was used to fit the encoding models but I can't find a discussion of regularization in the remainder of the paper. Was regularization actually used when fitting the encoding models? If not, what was the number of TRs used to fit the models and is overfitting not a concern? If yes, how were the regularization parameters chosen, and how does the regularization affect the feature importance estimates of the hidden units? I find the estimate of the timescale for each voxel a bit unintuitive (Equation 6). Can the authors give some intuition behind this formulation? I'm also not certain that the beta weights can be interpreted in the way that the authors do. Do hidden units with the same timescales really have very similar betas, for a specific voxel? Perhaps a convincing visualization would be a scatter plot of T_p vs. beta_p for all p, for a specific voxel. Visualizing this for voxels that appear red, blue, and white in Fig 3 would be ideal. L117: "unlike previous work, we use a stateful LSTM.." There is previous work that uses a stateful RNN (Wehbe et al. 2014 EMNLP) to model brain recordings, so this distinction doesn't seem necessary.

[Author Response · NeurIPS 2020]



Figure 1: Timescales estimated in MT model (revised after bug fix). Colormap follows Fig. 3 in main text.

We thank the reviewers for their insights and suggestions. Due to limited space, we will address more minor comments
in the camera-ready version should this paper be accepted. All references follow the main paper.

**Common queries: Voxel timescale estimate $T_v$ & negative $\beta$.** As noted in supplementary section 1.3, there was a
typo in the main text where Eq. 6 defines $T_v$. We use $\beta_p^2$, making $T_v$ agnostic to $\beta's$ sign. Further, the visualization
suggested by reviewer 3 (plotting $T_v$ vs. $\beta_p$) is shown in Supplementary Fig. 1B. This demonstrates the relationship
between $|\beta|$ and $T_v$, serving as a proof of concept. **Literature as ground truth/Inaccuracy of $\delta$-sum:** Firstly, section
4 demonstrates the inability of $\delta$-sum to down-sample long timescale representations, causing these features to be highly
correlated with word rate. We also find that AC voxels assigned *long* timescales by $\delta$-sum are, in fact, well predicted
by low-level models like word rate and acoustic spectrum, which change rapidly across time [4; Tang, Hamilton and
Chang, 2017]. Taken together, this evidence strongly suggests that long $T_v$ estimates by $\delta$-sum are false and caused
by the down-sampling confound. For precuneus and PFC, we find that $\delta$-sum, in contrast, assigns shorter timescales
than AC, going against the known language temporal hierarchy. **Leave-One-Out cross-validation for interpolation**
**weights $a$.** The model learns a weight $a_i$ on each word $w_i$ to interpolate word activations $W$ across time (Eq. 3). We
solve for $a$ in $a = \phi^{-1} W$ by ridge regression. The ridge coefficient is estimated by leaving one word out at a time and
measuring accuracy of interpolating its activation from other words. **Encoding model fits rely on cross-validation.**
To find regularization coefficients for $\beta$, we bootstrapped the regression procedure 50 times for each encoding model.
In each bootstrap, a random set of 5000 TRs (125 blocks of 40 consecutive TRs) were removed from training set (26
stories) and used as validation data. Ridge coefficients were picked based on the validation set's prediction performance,
averaged across bootstraps [4, 7]. Final model performance was computed on a separate test set (1 story).

**R#1: Merits of timescale estimator over previous methods.** To estimate timescale by manipulating context length,
separate encoding models are first built for each CL. A voxel's CL preference is then computed as the center of mass of
the *encoding performance curve* across different CLs. As discussed in the supplement, if the performance across CLs is
similar (curve is flat, Supp. Fig 1A), the voxel has a *large* center-of-mass, artificially inflating the CL preference. This
is the case with many AC voxels (Fig. 5). In comparison, our timescale estimation procedure is based on direct control
of timescales in LMs, and predicts short-timescales in primary AC. **Brain areas integrate speech over 14s.** Prior work
[2, 3, 8, 23] demonstrates through different experiments and methods that some brain regions integrate information over
long timescales, on the order of several seconds. **Transformer LMs:** The main contribution of this paper is to use LMs
with explicitly interpretable timescales to make detailed inferences about the brain. To the best of our knowledge, this is
currently lacking in Transformer-based LMs. While we observe slightly better encoding performance with Transformer
LMs (work under submission), to investigate the temporal hierarchy we are restricted to coarse CL preference estimates.
**Cross-subject consistency:** The histograms in Figs. 4-5 compare different timescale estimation procedures across
*all* 6 subjects. We found a bug in the colormap limits for subject S1, and show updated flatmaps in Fig. 1 here. The
patterns are highly similar across subjects, as are the drawbacks of the other methods shown in the supplementary
flatmaps. **Permutation test:** Block-wise permutation tests are entirely appropriate for assessing significance of model
predictions in this setting, and account for temporal autocorrelation. An average of 6.8% of voxels in a subject are
significant according to this test, demonstrating that non-permuted data doesn't always provide a better fit.

**R#2: Restricting CL:** The context length was restricted based on the back-propagation-through-time (BPTT) length in
the baseline model upon which the interpretable LM was based (Merity et al. [21]).

**R#3: Kernel choice:** In practice, many kernels could be used for interpolation. However, the RBF kernel 1) generalizes
to the $\delta$-sum method when $\epsilon \to \infty$ (this was a typo in the main text) and 2) has a kernel width that can be directly linked
to timescale. These properties are not exhibited by other commonly used alternatives, like polyharmonic spline kernels.
**LM Performance:** Perplexity on WikiText2 test set (lower is better): $68.33 \pm 0.12$. Baseline LSTM (no timescale
specification): $70.23 \pm 0.24$. These values are comparable to Merity et al. [21].

[Meta-Review · NeurIPS 2020]

The reviewers appreciated this paper's approach to explaining voxel activation as a function of time-lagged stimuli features. There were a few mistakes in the paper that were uncovered in the reviews, but adequately addressed in the author's rebuttal. The reviewers point out that the authors used a less-accurate LSTM model when more accurate models exist (e.g. transformers) but the authors point out that the time-lagged nature of the LSTM they use is crucial to their work. In general the reviewers were positive about this usage of a variant LSTM to explain the temporal nature of language processing in the brain.